# Oxidative Status, Iron Plasma Levels in Venous Thrombosis Patients

**DOI:** 10.3390/antiox13060689

**Published:** 2024-06-03

**Authors:** Salvatore Santo Signorelli, Andrea Barbagallo, Gea Oliveri Conti, Maria Fiore, Antonio Cristaldi, Margherita Ferrante

**Affiliations:** 1Department of Clinical and Experimental Medicine, University of Catania, Via Santa Sofia 80, 95123 Catania, Italy; 2Department of Medical, Surgical and Advanced Technologies “G.F. Ingrassia”, Hygiene and Public Health, University of Catania, Via Santa Sofia 73, 95123 Catania, Italy; olivericonti@unict.it (G.O.C.); mfiore@unict.it (M.F.); antonio.cristaldi@unict.it (A.C.);

**Keywords:** oxidative stress, venous thromboemebolism, metal, iron, deregulation

## Abstract

Exaggerated clot induces venous thrombosis (VTE); oxidative stress (OxS) can to be postulated as additional risk factor. This study evaluates firstly OxS by measuring surrogate biomarkers (malondialdehyde-MDA, 4-hydroxinonenal-4-HNE, superoxide desmutase enzyme (SOD)), secondly the iron (Fe) plasma level and thirdly the hepcidin protein (Hep) level in patients with VTE. A case control study was performed enrolling twenty hospitalized patients and an equal number of healthy individuals. In VTE patients, the following results were found. The MDA was 8.38 ± 0.5 µM/L, the 4-HNE measured 2.75 ± 0.03 µM/L and the SOD was 0.025 ± 0.01 U/mL. The I was 73.10 ± 10 µg/dL and the He was 4.77 ± 0.52 ng/mL. In the control group, the MDA measured 5.5 ± 0.6 µM/L, the 4-HNE 2.24 ± 0.021 µM/L and the SOD 0.08 ± 0.12 U/mL. The Hep was 2.1 ± 0.55 ng/mL and the Fe was 88.2 ± 9.19 µg/dL. Differences were statistically significant. Results suggest that in VTE there is activated OxS, Fe deregulation and over-production of Hep. Fe deregulation induces OxS, leading both to inflammation in the clot activator and stimulation of the pro-thrombotic status. The study highlights OxS and Fe and their regulation as intriguing indicators for risk of VTE.

## 1. Introduction

It is widely accepted that venous stasis, endothelial damage, hyper-clot (Virchow’s triad), congenital or acquired pro-thrombotic factors and concomitant diseases (i.e., heart failure, active cancer, stroke, lung disease, etc.) are fundamental in inducing venous thrombosis (VT) [1,2].

In addition, research has focused on the link between the oxidative stress induced by homeostasis transitional metals in inducing the pro-thrombotic condition [3,4,5]. Such environmental metals can generate oxidative stress (OxS), leading to the generation of reactive oxygen substances (ROS) and favoring the production of pro-oxidative agents (i.e., malondialdehyde, 4-hydroxynonenale). In addition, OxS causes the consumption of antioxidants (i.e., superoxide dismutase enzyme) [6,7,8].

Iron (Fe) is essential for cells whilst its excess or deficiency are dangerous. Interesting studies have demonstrated that diverse pathogenetic mechanisms with both high or low Fe plasma levels contribute to the risk of venous thromboembolism (VTE) [9,10,11,12]. This study firstly aims to evaluate activated OxS by measuring surrogate biomarkers (malondialdehyde-MDA, 4-hydroxinonenal 4-HNE, superoxide dismutase enzyme-SOD) in patients with VTE both for pulmonary embolism (PE) and deep vein thrombosis (DVT). Secondly, the Fe plasma level which is the third level of the hepcidin protein (Hep) is the most significant regulator key of Fe metabolism [12,13,14].

## 2. Study Methodolog

A case control study enrolled twenty patients from the Internal Medicine unit of the University Hospital ‘G. Rodolico’ (Catania, Italy) and an equal number of healthy controls to compare the findings. We did not consider patients with diabetes, active liver disease, stage III–V chronic renal insufficiency or active cancer. Eleven of the twenty VTE patients were men and seven were female, the mean age was 63.5 ± 2.2 years. Ten of twenty controls were men, and an equal number were female. The mean age was 55.8 ± 5.3 years (Table 1).

### 2.1. Determination of Surrogate Oxidate Biomarkers, Malondialdehyde-MDA and 4-Hydroxynonenal (4-HNE)

The methodology for determining both MDA and 4-HNE has already been described [6,15,16,17,18].

MDA and 4-HNE were determined as complexes with thiobarbituric acid (TBA) [15,17,18]. Plasma samples were treated as described above, with the exception of the incubation temperature which was 90.0 °C. A standard curve was generated using 1,1,3,3-tetraethoxypropane at five points as the external standard (Merck, Darmstadt, Germany), and the analytical recovery was calculated using MDA spiked blood samples with tetraethoxypropane standards with a mean recovery of 96%.

The 4-HNE was measured in 100 mL plasma (EDTA) using a reagent phase of TBA, HPLC-grade H_2_O and phosphoric acid (H_3_PO_4_, 0.15 M). The sample was incubated (45.8 °C for 1 h) and, when cold, centrifuged (15,000× *g*) for 10 min; then, it was filtered (syringe filter, 0.45 mm, Superchrom srl, Milan, Italy).

A total of 20 µL of the sample was then successively analyzed via HPLC (Perkin-Elmer) according to Signorelli’s study (2020) [18]. A four-point standard curve was generated using commercial 4-HNE from the Cayman Chemical Company (Ann Arbor, MI, USA). Mean recoveries of spiked matrices were 96 and 101% for MDA and 4-HNE, respectively. The MDLs were 0.02 μM/L for MDA and 0.01 μM/L for 4-HNE.

### 2.2. Superoxide Dismutase (SOD) Analysis

SOD was determined in plasma (EDTA) samples using a certificate assay kit from the Cayman Chemical Co. (Ann Arbor kits, MI 48108, USA). SOD activity was assessed by measuring the dismutation of superoxide radicals generated by xanthine oxidase and hypoxanthine in a 96-well plate. An SOD unit (U) is defined as the amount of enzyme required for 50% dismutation. In our study, cytosolic Cu-Zn-SOD was detected. Briefly, the plasma was centrifuged at 1500× *g* for 10 min at 4 °C. We collected only the top yellow plasma layer without disturbing the white buffer layer. The samples were diluted 1:5 with the sample buffer. Sample processing and plate development were carried out according to the manufacturer’s instructions. The plates were read at 440 nm through a Multiskan™ Thermo Fisher spectrophotometer (Thermo Fisher Scientific, Inc., Waltham, MA, USA) for 96-well plates. MDL was 0.005 U/mL.

### 2.3. Iron (Fe) Measurement

The Fe measurements were performed by the General Pathology Laboratory of the University Hospital G. Rodolico by using the methodology of SYNCHRON^®^ Systems FE/IBCT Beckman Coulter Ind. (Beckman Coulter Life Sciences, Cassin dei Pecchi, Italy). In brief, IBCT reagent is used to measure the iron concentration via a timed-endpoint method. In the reaction, iron is released from transferrin by acetic acid and is reduced to the ferrous state by hydroxylamine and thioglycolate. The ferrous ion is immediately complexed with the FerroZine Iron Reagent.

The SYNCHRON^®^ System(s) automatically proportions both the sample and reagent volumes into a cuvette. The automated system reveals the change in absorbance of samples at 560 nm. The change in absorbance is directly proportional to the concentration of iron bound to transferrin in the sample and is used by the system to calculate and express the total iron-binding capacity. The SYNCHRON^®^ System (Brooklyn, NY, USA) was calibrated using the SYNCHRON^®^ Systems FE/IBCT Calibrator Kit before the analysis of the samples.

### 2.4. Hepcidin (Hep) Measurement

Hep was measured using Elisa Hepcidin 25 (bioactive) HS ELISA RUO (EIA-5782R).

The He standards and controls were both reconstituted with deionized water and mixed several times.

Each standard control and sample was put into appropriate wells using an Eppendorf Research Plus multichannel micropipette (20 µL). After the addition of enzyme conjugate into each well, the microplate was mixed (10 s); then, it was incubated (60 min; 23 °C) and washed. All content of the microwells was fully removed using absorbent paper and, after adding the enzyme complex into all wells, the plate was incubated (30 min) at room temperature under shaking. After the second washing, 100 µL of substrate solution was added into all wells. Finally, after the last incubation (20 min; 23 °C), the enzymatic reaction was stopped by adding a stop solution.

The absorbance (OD) of each well at 450 ± 10 nm was determined to measure Hep (ng/mL) with a Multiskan™ Thermo Fisher spectrophotometer (Thermo Fisher Scientific, Inc., Waltham, MA, USA).

### 2.5. Statistical Analysis

Quantitative variables are presented as mean ± standard deviation, and these were compared with the Chi-square test. The serum plasma levels of the biomarkers were compared with the *t*-Student test. The level of statistical significance was set at *p*-value < 0.05.

## 3. Results

In VT patients, the MDA plasma level was 8.38 ± 0.5 µM/L, 4-HNE was 2.75 ± 0.03 µM/L and SOD was 0.025 ± 0.01 U/mL. The mean value of He was 4.77 ± 0.52 ng/mL. In the control group, we found MDA was 5.5 ± 0.6 µM/L, 4-HNE was 2.24 ± 0.021 µM/L and SOD was 0.08 ± 0.12 U/mL. Fe plasma levels were found in the VTE to be 73.10 ± 10.91 µg/dL and in controls to be 88.2 ± 9.19. Hep was 2.1 ± 0.55 ng/mL (Table 2).

## 4. Discussion

OxS is closely associated with the impaired balance of two systems (pro-oxidant and antioxidant) [17]. Antioxidant enzymes scavenge ROS to limit their detrimental effects [19]. ROS generation derives from the dysregulation of the redox mechanism which affects thrombotic imbalance leading to pathological consequences. Excessive ROS generation or a defect in the antioxidant defence system causes endothelial dysfunction, damage to the endothelial cell lining, activated coagulative factors and a shortfall in the anti-thrombotic and fibrinolytic systems [20]. A significant elevation in OxS (MDA, myeloperoxidase) surrogate biomarkers in patients affected with DVT was also reported. Reduced enzymatic antioxidant capability (SOD) was indicated as a significant agent in the development of thrombus in the venous circulation of the lower limbs [10]. It is known that the antioxidant enzymatic system is the first defence against oxidative injury [11].

Fe plays a role in enhancing coagulation, and in attenuating fibrinolysis [21]. Fe excess induces OxS, effecting clotting and tightening which can induce thrombus generation [9]. It may be interesting to investigate whether a beneficial effect of iron restriction on thrombosis inhibits OxS results. Iron restriction in diet accelerates venous thrombus resolution by inhibiting OxS [22] or by increasing intra-thrombotic neo-vascularization stress. OxS provokes cell damage (citotoxicity) and the expression of adhesion molecules, and it raises tissue factor expression, activates platelets and inhibits the anti-thrombotic capability of plasminogen. Interestingly, OxS damages endothelial cell properties, and reduces nitric oxide generation (NO). OxS generates the pro-coagulative condition and endothelial dysfunction and is dangerous to platelets.

Iron excess in endothelial cells provokes OxS and reduces NO availability, creating a deleterious pathway including endothelial dysfunction and platelet activation [23,24]. All these are crucial for the pro-thrombotic condition. Regarding the latter, the SOD enzyme contributes positively to the fluidity of the platelet membrane. This confirms that counteracting OxS by SOD enzyme activity against thrombus generation lowers VTE risk [19,23,24,25,26,27]. The excess of Fe favors the production of ROS through the oxidation of the ferrous form to the ferric through the Fenton reaction. The Fe reacts with the peroxides, leading to ROS generation. Consequently, there is an imbalance between oxidative and antioxidative agents leading to OxS. Results of our study show increased levels of MDA and 4-HNE, while SOD was reduced in VTE patients. The results seem to confirm the presence of the OxS pattern in patients with pro-thrombotic activated condition such as those suffering from VTE. We found differences in the Hep level between VTE and controls. In this regard, it is crucial to point out that Hep regulates I homeostasis. The Hep binds itself to the complexes ferroportin-ceruplasmin and ferroportin-efestine. It favors the internalization and degradation of the metal and it blocks the plasma extravasation of the metal, reducing the plasma level. Inflammation and high level of Fe increase Hep synthesis. We want to postulate a link between, on the one hand, the inflammatory pathway in VTE and, on the other hand, Fe and Hep. In this regard, it is known that such proinflammatory agents (IL 6, TNF) interact with the IL-6 receptor associated with 130 glycoprotein, inducing a cascade of signals. As a consequence, the enzyme JAK2 tyrosine chinase is activated, causing the fosforilation of several targets (i.e., the STAT). The STAT fosforilated through IL 6 causes the activation and translocation of the gene of the Hep. We have previously demonstrated an increase in the biomarkers of OxS in patients with VTE events. [5] The present paper aims to add additional data on the potential role of metal in provoking or favoring the risk of VTE. We accepted that Fe induces OxS, which is closely related to inflammation, which in turn causes an unbalancing of the coagulative homeostasis, favoring pro-coagulative (pro-thrombotic) conditions. The results seem to agree with previous studies; we want to suggest being aware both of OxS by measuring surrogate biomarkers and of Fe levels as interesting and helpful markers to evaluate VTE risk. 

## 5. Limits and Strengths

The small sample size must be considered as the major limit of our study.

However, we suggest drawing attention to the occurrence of oxidative stress in VTE patients. Concerning the latter, we want to note that oxidative stress favors the clot that is particularly involved as a pivotal mechanism for VTE. We enrolled hospitalized patients with VTE; thus, we believe we have provided true data originating in current medical practice. We consider our results to be in agreement with findings from previous studies and in addition we postulate that our results can add affirmative elements about additional mechanisms of the pathophysiology of thrombotic venous diseases. The oxidative stress originating in different agents (i.e., iron or metals) or mechanisms (i.e., hypoxia, blood stasis) plays a role in VTE and consequently biomarkers of the oxidative stress can be helpful in evaluating thrombotic risk.

## Figures and Tables

**Table 1 antioxidants-13-00689-t001:** Demographic characteristics of two enrolled groups.

	VTE	Controls	*p* Value
Age (years)	63.5 ± 2.2	55.8 ± 5.3	n.s.
Men (n.)	12	11	n.s.
Female (n.)	8	9	n.s.
Diabetes (n.)	1	1	n.s.
Arterial hypert. (n.)	3	2	n.s.
Heart failure (n.2)	2	2	n.s.
Kidney disease (n.2)	2	2	n.s.

The PE diagnosis was performed via computerized tomography and those diagnosed with PE underwent ultrasound examination of the venous circulation of the lower limbs. The non-compression of one or more of the deep veins of the lower limbs via the ultrasound probe (CUS test) was considered positive for DVT. All the patients were asked to give their informed consent to be enrolled in the study. Blood samples were then taken to measure for markers. n.s. = no significant.

**Table 2 antioxidants-13-00689-t002:** Plasma levels of the oxidative biomarkers MDA, 4-HNE, SOD, Fe and Hep. Results are expressed as mean value and standard deviation (*±*).

Biomarkers	VTE	Controls	*p* Value
MDA (µM/L)	8.38 ± 0.5	5.5 ± 0.6	<0.001
4-HNE (µM/L)	2.75 ± 0.03	2.24 ± 0.021	<0.05
SOD (U/mL)	0.025 ± 0.01	0.08 ± 0.01	<0.001
Hep (ng/mL)	4.77 ± 0.52	2.1 ± 0.55	<0.001
Fe (µg/dL)	73.10 ± 10.91	88.2 ± 9.19	<0.05

## Data Availability

The original contributions presented in the study are included in the article.

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
