# Peer review of "Oxidative Status, Iron Plasma Levels in Venous Thrombosis Patients"

_antioxidants, 2024, doi:10.3390/antiox13060689_

Round 1
Reviewer 1 Report
Comments and Suggestions for Authors
1. Please define inclusion/exclusion criteria clearly. Based on Table 1, your control group has diabetes, arterial hypertension, heart failure, and kidney disease?
2. Method section: Sections 2.1 and 2.3 need more details. Also, provide a description of the calibration of the equipment that you used.
3. The format for writing numbers should be corrected: change "2,1±0.55" to "2.1±0.55" and use a similar decimal format to show the values of similar parameters. Furthermore, remove additional points at the end of sub-titles. Moreover, change "Results (Table 2)" to "Results". Also, change title2 to Method section.
4. Why did you consider 0.05 for p-value? Why did you not use the Bonferroni-corrected p-value? Was there any dependency between the parameters?
5. In the first paragraph of the Discussion section, you should summarize your main findings, then proceed with your discussion as in the current version.
6. The most recent references are from 2020 and 2021. Please update the references with those from 2022 and 2023.
7. Add a brief conclusion section at the end of the manuscript.
8. I could not find any information about the IRB (Institutional Review Board) and ethical approval.
9. The English language level of the manuscript needs to be improved.
10. The clinical and medical novelty, as well as the practical application of your findings, are not clear to me. Who is the intended audience for this project, and how can a clinician practically apply these findings?
[https://doi.org/10.1186/s13104-019-4381-2]
Comments on the Quality of English LanguageThe English language level of the manuscript needs to be improved.
Author Response
Review. 1
Q1.Please define inclusion/exclusion criteria clearly. Based on Table 1, your control group has diabetes, arterial hypertension, heart failure, and kidney disease?
- Thanks for your question. I amended the manuscript including both inclusion and exclusion criteria (see page 2, lines 46 – 49).
Q 2. Method section: Sections 2.1 and 2.3 need more details. Also, provide a description of the calibration of the equipment that you used.
- I rewrote the 2 sections and answered the requested questions.
Q 3. The format for writing numbers should be corrected: change "2,1±0.55" to "2.1±0.55" and use a similar decimal format to show the values of similar parameters. Furthermore, remove additional points at the end of sub-titles. Moreover, change "Results (Table 2)" to "Results". Also, change title 2 to Method section.
- Thanks for your suggestion. Accordingly, I amended Table 2, changed its title and changed the title of section 2.
Q 4. Why did you consider 0.05 for p-value? Why did you not use the Bonferroni-corrected p-value? Was there any dependency between the parameters?
- Thanks for your question. There were 20 individuals for each group (patients vs. healthy) and we used the t-Student’s Test for paired data to compare the results from the 2 groups. The p-value level < 0.05 must be considered significant in comparing the findings
Q 5. In the first paragraph of the Discussion section, you should summarize your main findings, then proceed with your discussion as in the current version.
- Thanks for your suggestion. I briefly summarised the results of the study (see page 4, lines 138 – 140).
Q 6. The most recent references are from 2020 and 2021. Please update the references with those from 2022 and 2023.
- Thanks for your suggestion. Accordingly, in the reference list I listed the most recent papers (references 31, 32, 33).
Q 7. Add a brief conclusion section at the end of the manuscript.
- Thanks a lot for your suggestion. I modified lines 187 – 190 of page 5 to be considered as conclusive remarks
Q 8. I could not find any information about the IRB (Institutional Review Board) and ethical approval.
- Thanks for your question. I answered the requested concerns in the manuscript (see page 2, lines 49 – 51).
Q 9. The clinical and medical novelty, as well as the practical application of your findings, are not clear to me. Who is the intended audience for this project, and how can a clinician practically apply these findings?
- Thanks for your question. We demonstrated the role of oxidative stress in VTE by measuring the oxidative biomarkers [5]. It is known that an imbalance in iron favours oxidative stress. Furthermore, the role of oxidative stress in favouring clots was demonstrated. So, there is much backing for the role played by multiple agents in the pathophysiology of VTE. Heavy metals especially iron were found to be risk factors in VTE in specific disease settings (cancer, anaemia, active liver disease). Because iron induces oxidative stress closely linked to raised inflammation, its effect on coagulative homoeostasis favours the pro-coagulative (pro-thrombotic) condition. We feel that our results agree with previous research suggesting clinicians be aware of oxidative stress and iron levels in particular especially in specific pathological conditions (solid and haematological cancer, chemotherapy administration, anaemia, active liver disease). Measuring surrogate biomarkers and iron levels may help evaluate the potential or emerging risk of VTE.
Reviewer 2 Report
Comments and Suggestions for Authors
Review of the paper entitled „Oxidative status, iron plasma levels in venous thrombosis patients” by Signorelli Salvatore Santo, Barbagallo Andrea, Oliveri Conti Gea, Fiore Maria and Ferrante Margherita
My comments
The topic discussed by the Authors is interesting. Unfortunately, the paper is not well written and requires major improvement.
· Title
The title should be consistent with the content of the entire text and should indicate the most important result of the research conducted.
The authors titled their paper "Oxidative status, iron plasma levels in venous thrombosis patients". Meanwhile, the authors themselves wrote „This study firstly aims to evaluate activated OxS by measuring surrogate biomarkers (malondialdehyde, 4-hydroxynonenal, superoxide dismutase enzyme) in patients with VTE both for pulmonary embolism (PE) and deep vein thrombosis (DVT). Secondly, the iron plasma level which is the third level of the hepcidine enzyme (He) is the most significant regulator key of iron metabolism”. In my opinion, the current version's title is not consistent with the text of the paper. Moreover, „.....in venous thrombosis patients” is not correct. It should be in my opinion.... „in Patients with Venous Thromboembolism”.
· Abstract
The abstract should follow the style of structured abstracts, but without headings: 1) Background; 2) Methods; 3) Results: Summarize the article's main findings; and 4) Conclusion. In its current version, the Abstract presents almost exclusively detailed research results.
· Study metodology
A case control study enrolled twenty patients with venous thromboembolism (VTE). Venous thromboembolism is a complex disease combining deep vein thrombosis (DVT) and its most dangerous complication, pulmonary embolism (PE). Have D-dimer levels been measured in patients? Has the research been approved by the appropriate Ethics Committee? How was blood collected from patients (serum, citrate, EDTA or something else?).
The authors claim that they did not consider patients with diabetes, active liver disease, III – V stage of chronic renal insufficiency and active cancer. However, this information seems inconsistent with the data in the Table 1, which shows that there is 1 patient with diabetes and 2 patients with kidney diseases and from the control group there is 1 person with diabetes and 2 people with kidney diseases.
Results and Discussion
I propose to combine the "Results" section and "Discussion" section into one "Results and Discussion" section. In its current version, the Discussion is not well written. First of all, authors should discuss their results with those of other authors.
For example. Ekim et al indicated that the MDA level in the serum of DVT patients and control subjects is 0.13±0.06 µmol/L and 0.09±0.05 µmol/L, respectively [Ekim M, Sekeroglu MR, Balahoroglu R, Ozkol H, Ekim H. Roles of the Oxidative Stress and ADMA in the Development of Deep Venous Thrombosis. Biochem Res Int. 2014;2014:703128. doi: 10.1155/2014/703128. Epub 2014 Apr 13. PMID: 24818025; PMCID: PMC4003758].
Meanwhile, the serum level of MDA in the study and control groups shown by the authors is almost two orders of magnitude higher. Why? The authors should discuss this.
The authors should also cite their earlier paper [Signorelli SS, Conti GO, Fiore M, Elfio MG, Cristaldi A, Nicolosi I, Zuccarello P, Zanoli L, Gaudio A, Di Raimondo D, Ferrante M. Inter-Relationship between Platelet-Derived Microparticles and Oxidative Stress in Patients with Venous Thromboembolism. Antioxidants (Basel). 2020 Dec 2;9(12):1217. doi: 10.3390/antiox9121217. PMID: 33276677; PMCID: PMC7761576] which dealt with a similar problem, and compare the results obtained now with those that the authors received in the previous paper. There are also other studies. Therefore, it is not entirely true that the authors discussed the topic of oxidative stress markers in the course of VTE for the first time.
I was also interested in the results regarding iron and hepcidin. Hepcidin is a protein produced mainly in liver cells. It is the most important factor regulating iron management in the body. It inhibits the absorption of iron from the gastrointestinal tract and its release from macrophages, thereby reducing the concentration of iron in the blood serum. This means that the higher the hepcidin concentration, the lower the iron concentration and vice versa. Meanwhile, the authors' results indicate an inverse correlation. In the control group, the levels of hepcidin and iron were 4.77±0.52 ng/mL and 73.10±10.91 µg/dL, respectively, and in the patients group 2.1±0.55 ng/mL and 88.2±9.19 µg/dL, respectively. This would also need to be discussed. The authors should also compare their results with those of other authors. There are studies on this. For example: Ellingsen TS, Lappegård J, Ueland T, Aukrust P, Brækkan SK, Hansen JB. Plasma hepcidin is associated with future risk of venous thromboembolism. Blood Adv. 2018 Jun 12;2(11):1191-1197. doi: 10.1182/bloodadvances.2018018465. PMID: 29844204; PMCID: PMC5998931.
In short, there is little discussion of the actual data shown in this paper.
I believe the manuscript would benefit from a major revision in which the data are presented in a clearer way. A more detailed analysis and discussion of the actual results would also help.
I suggest also that English needs improvement.
Comments on the Quality of English Language
Numerous punctuation errors, typos, lack of consistency in the use of abbreviations. For example "desmutase enzyme". It should be "superoxide dismutase"; "......key of iron metabolism. [13,14]". It should be ".... key of iron metabolism [13,14]"; "Malondialdheyde". It should be "Malondialdehyde"; "Superoxide desmuate". It should be "Superoxide dismutase".
HNE, 4-HNE; VT, VTE
Author Response
Title
Q 1. The title should be consistent with the content of the entire text and should indicate the most important result of the research conducted. The authors titled their paper "Oxidative status, iron plasma levels in venous thrombosis patients". Meanwhile, the authors themselves wrote „This study firstly aims to evaluate activated OxS by measuring surrogate biomarkers (malondialdehyde, 4-hydroxynonenal, superoxide dismutase enzyme) in patients with VTE both for pulmonary embolism (PE) and deep vein thrombosis (DVT). Secondly, the iron plasma level which is the third level of the hepcidine enzyme (He) is the most significant regulator key of iron metabolism”. In my opinion, the current version's title is not consistent with the text of the paper. Moreover, „.....in venous thrombosis patients” is not correct. It should be in my opinion.... „in Patients with Venous Thromboembolism”.
- Thanks a lot for your suggestion. I have rewritten the title of the paper to: A Study on Oxidative Stress and Iron Metabolism in Patients with Venous Thromboembolism.
Abstract
Q 2.The abstract should follow the style of structured abstracts, but without headings: 1) Background; 2) Methods; 3) Results: Summarize the article's main findings; and 4) Conclusion. In its current version, the Abstract presents almost exclusively detailed research results.
- Thanks for your suggestion. Accordingly, I restructured the abstract including the suggested sections.
Study metodology
Q 3. A case control study enrolled twenty patients with venous thromboembolism (VTE). Venous thromboembolism is a complex disease combining deep vein thrombosis (DVT) and its most dangerous complication, pulmonary embolism (PE). Have D-dimer levels been measured in patientsHas the research been approved by the appropriate Ethics Committee? (Yes see before) How was blood collected from patients (serum, citrate, EDTA or something else?). (EDTA)
- Have D-dimer levels been measured in patients? Yes. All measures of the VTE patients were found higher compared to the laboratory control values.
Has the research been approved by the appropriate Ethics Committee?
- Thanks for your question. I added your concerns to the manuscript (page 2, lines 49 – 51).
The authors claim that they did not consider patients with diabetes, active liver disease, III – V stage of chronic renal insufficiency and active cancer. However, this information seems inconsistent with the data in the Table 1, which shows that there is 1 patient with diabetes and 2 patients with kidney diseases and from the control group there is 1 person with diabetes and 2 people with kidney diseases.
- I thank you for question. I amended the manuscript including both inclusion and exclusion criteria (see page 2, lines 46-49)
Results and Discussion
Q 4. I propose to combine the "Results" section and "Discussion" section into one "Results and Discussion" section. In its current version, the Discussion is not well written. First of all, authors should discuss their results with those of other authors.
For example. Ekim et al indicated that the MDA level in the serum of DVT patients and control subjects is 0.13±0.06 µmol/L and 0.09±0.05 µmol/L, respectively [Ekim M, Sekeroglu MR, Balahoroglu R, Ozkol H, Ekim H. Roles of the Oxidative Stress and ADMA in the Development of Deep Venous Thrombosis. Biochem Res Int. 2014;2014:703128. doi: 10.1155/2014/703128. Epub 2014 Apr 13. PMID: 24818025; PMCID: PMC4003758].
- Thanks for your suggestion. However, Ekim's study results were considered (page 4, lines 158 – 164. Dietary iron restriction accelerates venous thrombosis by inhibiting OxS or by increasing intra-thrombotic neo-vascularisation stress. OxS provokes cell damage (citotoxicity) and the expression of adhesion molecules, whereas it raises tissue factor expression, activates platelets, and inhibits the anti-thrombotic capability of plasminogen.) aimed at demonstrating the relationship between iron and oxidative stress and focusing on the effect of heavy metals on thrombotic condition outcomes.
Q 5.The authors should also cite their earlier paper [Signorelli SS, Conti GO, Fiore M, Elfio MG, Cristaldi A, Nicolosi I, Zuccarello P, Zanoli L, Gaudio A, Di Raimondo D, Ferrante M. Inter-Relationship between Platelet-Derived Microparticles and Oxidative Stress in Patients with Venous Thromboembolism. Antioxidants (Basel). 2020 Dec 2;9(12):1217. doi: 10.3390/antiox9121217. PMID: 33276677; PMCID: PMC7761576] which dealt with a similar problem, and compare the results obtained now with those that the authors received in the previous paper. There are also other studies. Therefore, it is not entirely true that the authors discussed the topic of oxidative stress markers in the course of VTE for the first time.
- I would like to underline that we believe we are contributing to the data on the potential role oxidative stress and metal can play in increasing the risk of venous thromboembolism (page 7, lines 197 – 199).
Q 6.I was also interested in the results regarding iron and hepcidin. Hepcidin is a protein produced mainly in liver cells. It is the most important factor regulating iron management in the body. It inhibits the absorption of iron from the gastrointestinal tract and its release from macrophages, thereby reducing the concentration of iron in the blood serum. This means that the higher the hepcidin concentration, the lower the iron concentration and vice versa. Meanwhile, the authors' results indicate an inverse correlation. In the control group, the levels of hepcidin and iron were 4.77±0.52 ng/mL and 73.10±10.91 µg/dL, respectively, and in the patients group 2.1±0.55 ng/mL and 88.2±9.19 µg/dL, respectively. This would also need to be discussed. The authors should also compare their results with those of other authors. There are studies on this. For example: Ellingsen TS, Lappegård J, Ueland T, Aukrust P, Brækkan SK, Hansen JB. Plasma hepcidin is associated with future risk of venous thromboembolism. Blood Adv. 2018 Jun 12;2(11):1191-1197. doi: 10.1182/bloodadvances.2018018465. PMID: 29844204; PMCID: PMC5998931.
- Thanks for your question. Hepcidin increase signals the internalisation of the ferroportin-hephastine/ceruloplasmin complex and therefore a reduction in plasma iron but it also leads to an accumulation of metal in the cell. Accumulation in turn causes cell oxidative stress directly aggravated by the inflammation.
Q 7.The authors should also compare their results with those of other authors. There are studies on this. For example: Ellingsen TS.
- In my opinion, our study data does not agree with the concluding remarks of TS Ellingsen's
Reviewer 3 Report
Comments and Suggestions for Authors
The manuscript's authors aim to evaluate activated oxidative stress by measuring surrogate biomarkers in patients with venous thromboembolism both for pulmonary embolism and deep vein thrombosis. Venous thrombosis is one of the most common diseases affecting the conditions of blood circulation, and as a result, deepening the understanding of the backgrounds of this phenomenon has significant scientific and clinical significance. Since oxidative stress is involved in all the main processes in developing venous thrombosis, the issue raised by the manuscript's authors may interest a wide range of readers. The authors suggest that their findings may increase understanding of the pathophysiology of thrombotic venous diseases and help assess thrombotic risk. However, it would be advisable for the authors to describe in more detail the clinical significance of their studies in the Discussion section.
Authors should present their findings concisely in a separate section (Conclusion) of the manuscript. The references provided by the authors are relevant, but the latest relevant publications must also be included in the list of references.
In addition
1. In the text, there is no information about the presence of Helsinki Committee permission to conduct the described studies.
2. It is necessary to add information about the used blood fence and its separation to the text. The authors do not indicate which anticoagulants were used.
3. In the section Statistical analysis, it does not indicate which program was used by the authors.
Author Response
The manuscript's authors aim to evaluate activated oxidative stress by measuring surrogate biomarkers in patients with venous thromboembolism both for pulmonary embolism and deep vein thrombosis. Venous thrombosis is one of the most common diseases affecting the conditions of blood circulation, and as a result, deepening the understanding of the backgrounds of this phenomenon has significant scientific and clinical significance. Since oxidative stress is involved in all the main processes in developing venous thrombosis, the issue raised by the manuscript's authors may interest a wide range of readers. The authors suggest that their findings may increase understanding of the pathophysiology of thrombotic venous diseases and help assess thrombotic risk. However, it would be advisable for the authors to describe in more detail the clinical significance of their studies in the Discussion section.
Q 1. Authors should present their findings concisely in a separate section (Conclusion) of the manuscript. The references provided by the authors are relevant, but the latest relevant publications must also be included in the list of references.
- Thanks for your suggestion. I have revised the reference list and included some very recent publications.
In addition
Q 2. In the text, there is no information about the presence of Helsinki Committee permission to conduct the described studies.
- I thank you for suggestion. I emended the text (see page 2, lines 49-52)
Q 3. It is necessary to add information about the used blood fence and its separation to the text. The authors do not indicate which anticoagulants were used.
- Blood samples were drawn in EDTA anticoagulated tubes.
Q 4. In the section Statistical analysis, it does not indicate which program was used by the authors.
R.Thanks for your suggestion. I included the statistical program (see page 3, lines 131 – 133) in the text.
Round 2
Reviewer 2 Report
Comments and Suggestions for Authors
The manuscript has been substantially revised. The authors took my comments into account. However, the manuscript still needs a thorough improvement of the English language.
Dear Editor,
In my previous review, I also suggested improving the English language due to numerous punctuation errors, typos, etc.
Some examples:
It is: Study on oxidative stress and iron metabolism in patients with the Venous Thromboembolism.
It should be: Study on oxidative stress and iron metabolism in patients with the Venous Thromboembolism
It is: 2,75±0.03
It should be: 2.75±0.03
It is: 4-hydroxinonenale
It should be: 4-hydroxynonenal
It is: ...... are fundamental in inducing venous thrombosis (VT). [1,2]
It should be: .....are fundamental in inducing venous thrombosis (VT) [1,2].
It is: ethylenediamine tetra-acetic acid
It should be: ethylenediaminetetraacetic acid
It is: Malondialdheyde
It should be: Malondialdehyde
Best regards,
Anna Bilska-Wilkosz
Author Response
Q.1 There are numerous typos and grammar mistakes throughout, for example, the spellings for MDA, 4-HNE, hepcidin (hepcidin is a peptide hormone, not hepcidine enzyme?). In addition, the methods section for hepcidin measurement
are poorly written- it appears as a protocol from the manufacturer.
R. I emended both typos and grammar mistakes of the manuscript. According to the suggestion I rewritted the section for the hepcidne measuremt.
Q.2 The scientific units for MDA or 4-HNE are not correct. The units in the text do not correlate with the data in Table 1.
R. I emended the units concerning the biomarkers (MDA,4-HNE), and I revised the table 1
